# The Emergence and Dissemination of Multidrug Resistant *Pseudomonas aeruginosa* in Lebanon: Current Status and Challenges during the Economic Crisis

**DOI:** 10.3390/antibiotics11050687

**Published:** 2022-05-19

**Authors:** Ahmad Ayman Dabbousi, Fouad Dabboussi, Monzer Hamze, Marwan Osman, Issmat I. Kassem

**Affiliations:** 1Faculty of Medicine, Beirut Arab University, Beirut 11072809, Lebanon; daboussiahmad@gmail.com; 2Laboratoire Microbiologie Santé et Environnement (LMSE), Doctoral School of Sciences and Technology, Faculty of Public Health, Lebanese University, Tripoli 1300, Lebanon; fdabboussi@ul.edu.lb (F.D.); mhamze@ul.edu.lb (M.H.); 3Department of Public and Ecosystem Health, College of Veterinary Medicine, Cornell University, Ithaca, NY 14853, USA; 4Center for Food Safety, Department of Food Science and Technology, University of Georgia, 1109 Experiment Street, Griffin, GA 30223-1797, USA

**Keywords:** *Pseudomonas aeruginosa*, MDR, carbapenemase, Lebanon

## Abstract

*Pseudomonas aeruginosa* is a common cause of healthcare-associated infections and chronic airway diseases in non-clinical settings. *P. aeruginosa* is intrinsically resistant to a variety of antimicrobials and has the ability to acquire resistance to others, causing increasingly recalcitrant infections and elevating public health concerns. We reviewed the literature on multidrug-resistant (MDR) *P. aeruginosa* isolated from humans (nosocomial and community-associated), animals, and the environment in Lebanon, a country that has been suffering from a surge in antimicrobial resistance (AMR). We identified 24 studies that described the epidemiology and antimicrobial susceptibility profiles of *P. aeruginosa*. Our analysis showed that the bacterium was predominant in lesions of patients on mechanical ventilation and in burn patients and those with diabetic foot infections and hematological malignancies. We also found that carbapenem resistance in *P. aeruginosa* isolates in Lebanon involved both enzymatic and non-enzymatic mechanisms but depended predominantly on VIM-2 production (40.7%). Additionally, MDR *P. aeruginosa* was detected in animals, where a recent study reported the emergence of carbapenemase-producing *P. aeruginosa* in livestock in Lebanon. Notably, no studies evaluated the contribution of MDR *P. aeruginosa* in the environment to human infections. Taken together, our findings highlight the need for AMR surveillance programs and a national action plan to combat resistance in Lebanon.

## 1. Introduction

*Pseudomonas aeruginosa* is a known bacterial pathogen that can cause severe infections in humans. *P. aeruginosa* is also notable for its intrinsic and acquired resistance to a broad spectrum of clinically-relevant antibiotics. The prevalence of community- and healthcare-associated multidrug-resistant (MDR) *P. aeruginosa* infections has been increasing over the last few decades, becoming a global public health threat and leading to difficult-to-treat infections [1]. Subsequently, the World Health Organization (WHO) has classified *P. aeruginosa* among the prominent drug-resistant bacteria that require urgent research and the development of effective therapeutic interventions [2].

Despite the importance of *P. aeruginosa*, the prevalence and spread of MDR strains of this opportunistic pathogen have not been fully investigated in Lebanon. This is especially important, because AMR has been steadily increasing and potentially causing significant morbidity and mortality in Lebanon [3,4,5,6]. Furthermore, limited studies on AMR in Lebanon have revealed insufficient knowledge and inadequate practices related to antibiotics and AMR in general, including weak stewardship in clinical settings, over-the-counter access to antimicrobials in community pharmacies and a lack of awareness to the dangers of inappropriate use in the Lebanese community at large [7,8]. Additionally, the healthcare system in Lebanon has been affected by many emerging and significant challenges, including a collapsing national economy, the COVID-19 pandemic, and the burden of providing services for ~1.5 million Syrian refugees [9]. The devaluation of the Lebanese currency and the significant decrease in the quality of life have resulted in a serious shortage in healthcare professionals, including infectious diseases specialists and clinical microbiologists due to immigration. Consequently, the United Nations Economic and Social Commission for Western Asia (ESCWA) reported that more than 82% of the Lebanese population has limited access to healthcare services [10].

The economic crisis has also affected drug availability and in vitro diagnostics for various medical applications, including screening for antimicrobial resistance, while AMR surveillance and research programs (in clinical settings and the environment) became difficult to conduct [9,11]. For example, the American University of Beirut Medical Center (AUBMC), one of Lebanon’s largest tertiary care hospitals, has been running perilously short of medical supplies, affecting more than 500 critical items. The hospital now rations antibiotics. As a result, many patients are delaying treatment, leading to a higher risk of invasive infections and prolonged hospital stays [12]. This is not surprising when considering that Lebanon relies heavily on imports to meet its needs of medical and research supplies, and the severe devaluation of the Lebanese currency has limited purchases from foreign suppliers [9]. Predictably, the repercussions of a failing economy extend beyond the medical sector and impact agricultural practices and the environment, resulting in widespread pollution of critical resources in the country [11]. For example, prior to the crisis, Lebanon was already witnessing an emergence of resistance to last resort antibiotics in food industries, poultry farms, aquaculture, and surface water (irrigation, rivers, and sea), partially due to a debilitated infrastructure and suboptimal agricultural practices [5,6,13,14,15,16,17,18,19,20]. Therefore, AMR has been amplified due to many interacting reasons that include a weakened ability to tackle domestic and agricultural waste and maintain good animal farming practices. Taken together, the aforementioned observations suggest that the emergence of AMR and complicated infectious diseases in Lebanon will increase further.

In response to elevated concerns about AMR and infectious diseases in Lebanon, this review aimed to gather existing evidence on antibiotic-resistant *P. aeruginosa* in order to address the knowledge gaps regarding its epidemiology in Lebanon, a developing East Mediterranean country that is facing unprecedented economic and healthcare challenges.

## 2. Mechanisms of Antimicrobial Resistance in *Pseudomonas aeruginosa*

*P. aeruginosa* is resistant to numerous antimicrobials, which is facilitated by different mechanisms that include (1) restricting outer membrane permeability, (2) the expression of many efflux systems (e.g., constitutive MexAB-OprM, inducible MexXY-OprM), (3) the production of naturally-occurring antimicrobial-inactivating enzymes such as the hydrolytic β-lactamase enzymes (*bla*_AmpC_ and *bla*_OXA-50_) and the aminoglycoside modifying enzyme (AME) APH(3′)-IIb, and (4) mutations and enzymes that modify the targets of the antimicrobials (Figure 1) [21,22]. *P. aeruginosa* is also capable of developing antimicrobial resistance via horizontal gene transfer and the acquisition of resistance genes. It is important to note that both intrinsic and acquired resistance mechanisms play an important role in the evolution of MDR *P. aeruginosa*. For example, carbapenem resistance mechanisms in *P. aeruginosa* include the overexpression of AmpC enzyme, the acquisition of extended-spectrum β-lactamase (ESBL) and/or carbapenemase encoding genes through horizontal gene transfer, reduction in membrane permeability (e.g., mutations in the outer membrane porin, OprD), overexpression of *mexAB-oprM* efflux pump, and/or modification of penicillin binding proteins (PBPs) [23,24,25].

Similarly, aminoglycosides resistance in *P. aeruginosa* has been associated with an array of resistance mechanisms. For example, *P. aeruginosa* can inactivate aminoglycosides using AMEs which include acetyltransferases (AAC), nucleotidyltransferases (ANT), and phosphotransferases (APH). Numerous AMEs that were reported in the literature confer resistance to different aminoglycosides. Specifically, (i) AAC(3)-X, AAC(6′)-Ib, ANT(4′)-I, ANT(4′)-II, APH(3′)-IIIa, and APH(3′)-VIb display resistance against amikacin, (ii) AAC(3), AAC(6′)-I, AAC(6′)-II, and ANT(2″) confer resistance to gentamicin, and (iii) AAC(3)-II, ANT(2″), ANT(4′)-I, and ANT(4′)-II inactivate tobramycin. Resistance to aminoglycosides also includes overexpression of efflux pumps (particularly the MexXY-OprM complex), modification of 16S ribosomal RNA by methylases (e.g., *rmtA* and *rmtB*; preventing aminoglycosides from effectively binding to ribosomes), and decreased permeability [26].

Notably, *P. aeruginosa* can potentially develop resistance to fluoroquinolones, colistin, and fosfomycin. For example, although fluoroquinolones are frequently used to control infections with this bacterium, mutations in quinolone-resistance associated genes (i.e., *gyrA*, *gyrB*, *parC* and *parE*) along with the overexpression of resistance–nodulation–division efflux pumps (i.e., MexAB-OprM, MexCD-OprJ, MexEF-OprN, and MexXY-OprM) [27] are prevalent determinants that contribute to fluoroquinolone resistance in *P. aeruginosa*.

Colistin (polymyxin E) has been used as a last resort for the treatment of MDR and extensively drug resistant (XDR) *P. aeruginosa* infections [28]. However, in recent years, resistance to colistin has emerged around the globe, complicating the clinical management of certain MDR infections [29]. The drug’s consumption in Lebanese hospitals has increased 5× between 2010 and 2017; highlighting the importance of this drug in treating a variety of recalcitrant infections as well as the increasing number of complicated and MDR infections in Lebanon [19]. Resistance to colistin is commonly facilitated by mutations in genes associated with the modification of the lipid A of LPS and/or through the acquisition of the mobile colistin resistant (*mcr*) genes [30]. The emergence, spread, and notable transmissibility of *mcr* have raised public health concerns over the loss of the efficacy of colistin in treating MDR *P. aeruginosa* and other bacterial pathogens. Although different *mcr* genes have been widely reported in *Enterobacterales*, only *mcr-1* [31,32,33] and *mcr-5* [34] have been identified in sporadic *P. aeruginosa* isolates so far.

Due to the limited use of and the relatively low level of reported fosfomycin resistance in *P. aeruginosa*, this drug has been revisited to control antimicrobial-resistant *P. aeruginosa* strains [35]. Similarly to the antimicrobials discussed above, resistance to fosfomycin in *P. aeruginosa* can develop and is mainly associated with the gene encoding the inactivating enzyme, FosA [36], or the inactivation of the fosfomycin transport protein (GlpT) [37].

Taken together, it is perhaps clear that the antimicrobial options for treating and controlling *P. aeruginosa* infections are becoming increasingly limited, mainly due to the ability of this bacterium to develop resistance. This is predicted to have a serious impact on the emergence and spread of *P. aeruginosa* infections, especially in resource-limited countries like Lebanon.

## 3. Epidemiology of *Pseudomonas aeruginosa* Resistance in Lebanon

It is important to assess the epidemiology and the molecular mechanisms of resistance of MDR *P. aeruginosa* in Lebanon in order to guide empirical treatment choices. However, information on AMR in general and MDR *P. aeruginosa* in particular is lacking. For this purpose, we screened the literature to provide accessible and science-based evidence on the scope of this problem in Lebanon. Consequently, PubMed, Science Direct, Scopus, and the Google Scholar databases were mined for epidemiological studies on *P. aeruginosa* in hospital and/or extra-hospital settings that were published up to December 2021. We used a combination of the following terms: “*Pseudomonas*”, “*aeruginosa*”, “Lebanon”, “Susceptibility”, “Resistance”, “Antimicrobial”, “Antibiotic”, “AMR”, “Epidemiology”, “Imipenem”, “Meropenem”, and “Carbapenem”. Indexed original articles in English and French of any epidemiological design and sampling strategy and of any enrollment timing (retrospective, prospective, or cross-sectional) were included. Other types of reports, such as case reports, case series, and narrative and systematic reviews were excluded (Figure 2). Studies were eligible for inclusion in the review if they reported original information regarding the epidemiology of *P. aeruginosa* and its resistance to antibiotics in Lebanon. After importation of the search results, two authors (A.A. Dabbousi and F. Dabboussi) independently screened the citations for their relevance using the title and abstract and all qualified citations were retained for full-text assessment to confirm eligibility. Backward reference screening was done for all articles. Data extraction was performed by the same authors through a format prepared on a Microsoft Excel workbook.

The search strategy initially resulted in 1230 studies. Subsequently, a total of 40 manuscripts were screened in the full-text review and 24 studies describing the epidemiology and susceptibility profiles of *P. aeruginosa* isolates in Lebanon were identified as eligible according to our inclusion criteria (Table 1). The number of studies excluded or included at each stage is summarized in Figure 2. There were three nationwide studies identified; however, most of the included studies were conducted in Beirut (12 of 21), particularly at the American University of Beirut Medical Center (AUBMC) (N = 5). Additionally, eight studies were performed in the North and Akkar governorates (North Lebanon region). Furthermore, almost all reports (23 of 24) studied human samples, particularly in hospital settings. Only one paper investigated the susceptibility patterns of *P. aeruginosa* in animals, reporting the emergence of *P. aeruginosa* producing VIM-2 carbapenemase in Lebanese livestock [38].

The first data on AMR profiles of *P. aeruginosa* in Lebanon were generated at the AUBMC and Makassed general hospital located in Beirut [42,43]. At that time, 11% of the circulating clinical *P. aeruginosa* isolates were resistant to ceftazidime [42,43] and 8% were resistant to imipenem [43]. In the first decade of the 21st century, it appears that the susceptibility of strains to ceftazidime and imipenem significantly decreased. Specifically, in a 11-year retrospective study at the AUBMC, the prevalence of resistance against both antimicrobials reached 17% and 19%, respectively [44]. However, another large scale study that included 5090 clinical samples from patients suffering from healthcare- or community-acquired infections at the Hôtel-Dieu de France Hospital (Beirut) showed a higher prevalence of resistance to ceftazidime (34.7%) and imipenem (41.1%) [47]. Similar findings were obtained in other Lebanese geographical regions such as Tripoli, the North governorate of Lebanon [54]. Recent data from three studies conducted between 2014 and 2018 in local tertiary care centers in Beirut, North, and Akkar governorates were alarming, showing that 40–97.1% of *P. aeruginosa* infections were due to carbapenem-resistant isolates [51,56,58]. Furthermore, resistance against other antimicrobials has been reported in the Lebanese clinical settings. For example, 26.4% and 36.2% of the isolates were resistant to amikacin and levofloxacin between 2005 and 2009 at Hôtel-Dieu de France Hospital, respectively [47]. Yet again, the susceptibility of circulating *P. aeruginosa* against different antimicrobials appears to have continued to decrease over time. Although *P. aeruginosa* susceptibility patterns have been reported in several studies in Lebanon during the last three decades, most reports investigated a limited number of isolates (<150 isolates in 16 studies) in monocentric study locations. Therefore, like other clinically important pathogens, the full burden of AMR of *P. aeruginosa* in Lebanon remains unclear due to the lack of national surveillance data, a limited number of well-designed national studies, weak epidemiological tracking, and the absence of adequate funding, infrastructure, and oversight among other factors [9]. Nevertheless, the three nationwide retrospective investigations based on aggregated institutional antimicrobial susceptibility testing data from tertiary care centers located in different Lebanese districts have confirmed the relatively high level of resistance to ceftazidime, imipenem, amikacin, and levofloxacin. This was corroborated by the results observed in local studies conducted in Lebanon [39,40,41]. Additionally, a nationwide study reported for the first time the emergence of colistin-resistant *P. aeruginosa* isolates in Lebanon [39]. This was followed by another more recent and geographically-constrained study that also corroborated the emergence of colistin-resistant *P. aeruginosa* isolates in Lebanese clinical settings [51]. Taken together, these findings highlight a worrying trend that has been developing in Lebanon in recent decades and perhaps reflect the inappropriate use and/or over-reliance on carbapenems and colistin in the treatment of infections [7,9].

Unfortunately, only a few studies have addressed the occurrence of resistance genes in *P. aeruginosa* (Figure 3). Nevertheless, the limited data showed that carbapenem-resistant *P. aeruginosa* in Lebanon encompassed (1) carbapenem-hydrolyzing enzymes (including *bla*_VIM-2_, *bla*_GES-6_, *bla*_IMP-1_, *bla*_IMP-2_, and *bla*_IMP-15_), (2) non-enzymatic mechanisms (alteration of the outer membrane porin protein OprD, overexpression of efflux pumps), and (3) a combination of reduced membrane permeability and/or drug efflux pumps with enzyme inactivation mechanisms such as Class C β-lactamase hyperproduction (e.g., PDC-13, AmpC) [41,50,52,53,60]. It should be noted that a fingerprint analysis of strains isolated from various Lebanese hospitals indicated that the VIM-2 occurrence in *P. aeruginosa* was primarily due to clonal dissemination [41]. These results corroborated previous reports in the Middle East and North Africa (MENA) region that showed that different types of carbapenemases (VIM-2 was predominant) have been described in *P. aeruginosa* isolates in the countries surrounding Lebanon [60]. Recently, the emergence of carbapenemase-producing *P. aeruginosa* harboring *bla*_VIM-2_ has also been reported in livestock in Lebanon, potentially suggesting a zoo-anthropogenic transmission of VIM-2 producing *P. aeruginosa* and raising further concerns about the dissemination of MDR *P. aeruginosa* in animals and via zoonosis [55].

Due to the emergence and spread of resistance to traditional antimicrobial agents, healthcare professionals have been using the U.S. Food and Drug Administration (FDA) approved antipseudomonal beta-lactam drugs, ceftolozane/tazobactam and ceftazidime/avibactam [62]. These two new combinations of β-lactam/β-lactamase inhibitor antibiotics were recently registered at the Lebanese Ministry of Public Health as agents active against many MDR isolates of *P. aeruginosa* [62,63]. Regarding ceftolozane/tazobactam use in Lebanon, a study performed at AUBMC has shown a high susceptibility (96%) against non-MDR *P. aeruginosa* isolates but a low susceptibility (42%) against MDR isolates [63]. To date, national data on the resistance of *P. aeruginosa* to ceftazidime/avibactam are not available. These observations further highlight the urgency needed to tackle the glaring gaps in knowledge about MDR *P. aeruginosa* infections and associated controls and treatments in Lebanon.

## 4. Conclusions

Although Lebanon joined the World Health Organization’s (WHO) Global Antimicrobial Resistance Surveillance System (GLASS) in 2017, antimicrobial stewardship is still underdeveloped across the country. This situation has likely resulted in MDR *P. aeruginosa* to be prevalent in Lebanese hospitals and precipitated the emergence of carbapenem resistance that is associated predominantly with VIM-2 production. In conclusion, there is a critical need to establish robust monitoring and AMR stewardship programs and to devise interventions at the policy level that will bolster a national strategic plan to combat AMR in Lebanon. Otherwise, the country will face undesirable public health problems. AMR stewardship programs and extensive awareness campaigns must be integrated in the Lebanese vulnerable health system which has been impeded by a plethora of challenges. These interventions are required to curb mortality and morbidity due to AMR in the Lebanese population as well as in the large refugee population that is currently hosted in Lebanon. Given the proximity of Lebanon to many European, Middle Eastern and African countries and the mobility of the Lebanese and refugee populations, there is a risk that MDR can spill across the Lebanese borders, affecting other countries in the region and beyond.

## Figures and Tables

**Figure 1 antibiotics-11-00687-f001:**
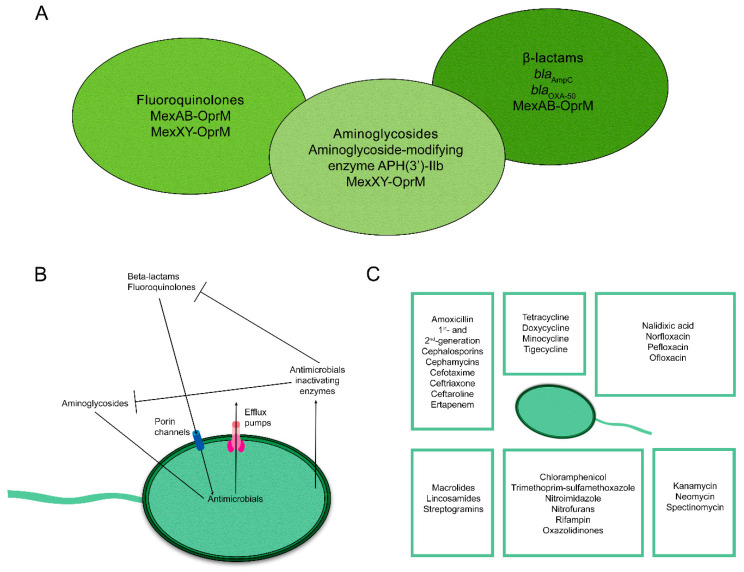
Antimicrobial resistance in *Pseudomonas aeruginosa*. Mechanisms of intrinsic and acquired AMR in *Pseudomonas aeruginosa* (**A**,**B**). The mechanisms include restricted outer-membrane permeability, efflux systems (MexAB-OprM and MexXY-OprM), and the production of antibiotic-inactivating enzymes (*bla*_AmpC_ and *bla*_OXA-50_). A list of the antimicrobials that are ineffective in the treatment of *P. aeruginosa* infections (**C**).

**Figure 2 antibiotics-11-00687-f002:**
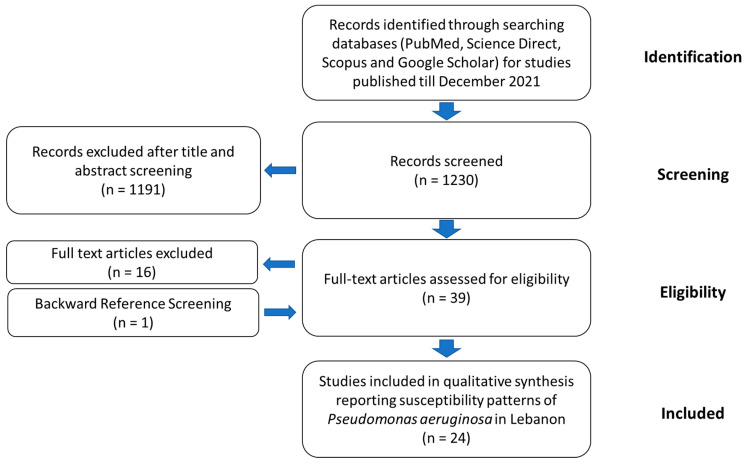
A Flow diagram describing the selection of the studies and the inclusion/exclusion process for the review according to PRISMA guidelines.

**Figure 3 antibiotics-11-00687-f003:**
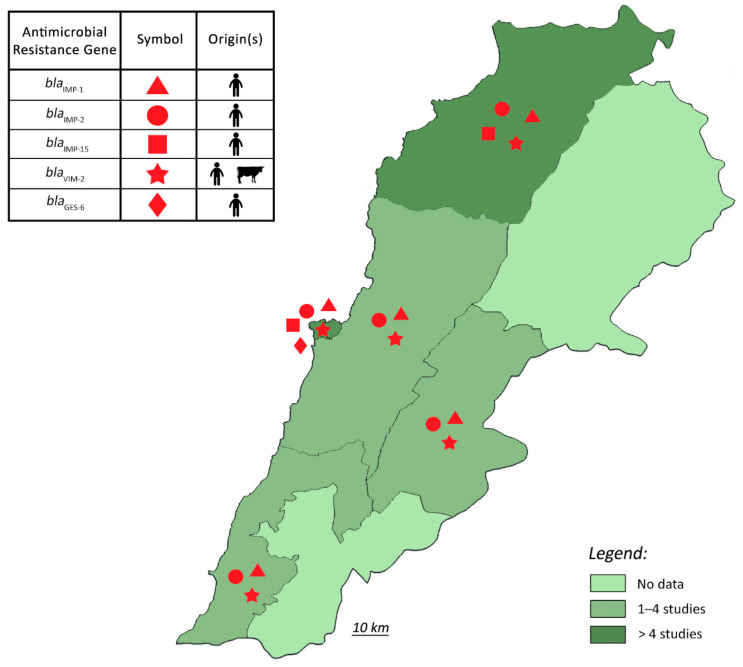
The geographical distribution of β-lactamase genes in *Pseudomonas aeruginosa* isolated from humans and animals in Lebanon.

**Table 1 antibiotics-11-00687-t001:** A list of the studies that described the epidemiology of antimicrobial resistant *Pseudomonas aeruginosa* in Lebanon.

Study Period	Study Design	Investigation Location	Sample/Isolate Type (N)	Special Resistance Phenotype	No. of CRPa Isolates (%)	Antimicrobial Resistance Rate (%)					β-Lactam Resistance Genes (N)	Ref.
						CAZ	IMP ^!^	CST	AMK	LVX ^§^		
**Nationwide studies**												
2015–2016	Retrospective	13 different hospitals	Clinical isolates (9005)	RI	2701 (30)	20	30	2	15	27	-	[39]
2011–2013	Retrospective	16 different hospitals	Clinical isolates (7897)	RI	2148 (27.2)	18.15	27.2	-	11.1	22.7	-	[40]
2012	Retrospective	11 different hospitals	Clinical isolates (1571)	RI	679 (43%)	-	43	-	-	-	*bla*_IMP-1_ (3) *bla*_IMP-2_ (7) *bla*_VIM-2_ (29)	[41]
**Beirut**												
1989–1996	Cross-sectional	Makassed General Hospital	Clinical isolates (2387)	RI	-	11.5	-	-	11	-	-	[42]
1992–1993	Cross-sectional	American University of Beirut Medical Center (AUBMC)	Blood (15) RT (264) Urine (176) Wound (163) Others (43)	RI	53 (8)	11	8	-	5	-	-	[43]
2000–2011	Retrospective	AUBMC	Blood (120) RT (1104) Urine (600) Wound (504) Others (72)	RI	456 (19)	17	19	-	15	21	-	[44]
2001	Prospective	AUBMC	LRT (8)	RI	3 (37)	-	37	0	0	0	-	[45]
2003–2004	Cross-sectional	AUBMC	Nosocomial (90) Hospital environment (18)	RI	6 (6.7)	4.5	6.7	-	1.11	3.3	-	[46]
2005–2009	Retrospective	Hôtel-Dieu de France Hospital	Nosocomial (4198) Community (892)	RI	2093 (41.1)	34.7	41.1	-	26.4	36.2	-	[47]
2006–2008	Retrospective	Lebanese Hospital Center (Beirut)	Nosocomial (25)	RI	17 (68)	-	68	-	0	-	-	[48]
2008–2017	Retrospective	AUBMC	Diabetic foot (34)	RI	1 (3)	8	3	-	-	18	-	[49]
2011–2012	Cross-sectional	Hôtel-Dieu de France Hospital	Clinical isolates (115)	CRPa	115 (100)	100	100	-	100	100	*bla*_IMP-1_ (2) *bla*_IMP-2_ (2) *bla*_VIM-2_ (18)	[50]
2014–2018	Retrospective	Lebanese Geitaoui Hospital	Blood (6) LRT (6) Urine (2) Wound (45)	RI	141 (54)	46.2	54	5.1	42	28.9	-	[51]
2015	Cross-sectional	Hospital of Saint Joseph of the Sisters of the Holy Cross	Nosocomial UTIs (12)	CRPa	12 (100)	100	100	0	100	100	*bla*_GES-6_ (9) *bla*_IMP-15_ (2) *bla*_VIM-2_ (6) Mutation in the *oprD* gene (8)	[52]
2016–2017	Cross-sectional	Saint George Hospital University Medical Center	Rectal swabs from ICU patients treated with carbapenem for >1 week (4)	CRPa	4 (100)	100	100	0	100	100	*bla*_VIM-2_ (3) Mutation in the *oprD* gene (4)	[53]
**North and Akkar**												
1998–2001	Cross-sectional	Islamic Hospital	Blood (9) ENT (77) Respiratory (27) Urine (180) Wound (99) Others (72)	RI	120 (25.9)	36.5	25.9	0	31	42.5	-	[54]
2006–2013	Retrospective	Nini Hospital	Clinical isolates (35)	CRPa	34 (97.1)	-	97.1	-	-	-	*bla*_IMP-15_ (2) *bla*_VIM-2_ (16) Mutation in the *oprD* gene (35) AmpC hyperproduction (8) Efflux pump (34)	[55]
2009–2015	Cross-sectional	Nini Hospital	ENT (136)	RI	9 (6.6)	8.1	6.6	0	10.3	10.3	-	[56]
2010–2011	Cross-sectional	Nini Hospital	ENT (8) LRT (30) Urine (23) Wound (20) Others (7)	RI	19 (21.6)	22.6	21.6	-	-	-	-	[57]
2015–2017	Cross-sectional	El Youssef Hospital Center	Urine (45)	RI	404 (40)	28.9	40	0	6.7	46.7	-	[58]
2015–2017	Cross-sectional	Nini Hospital and El Youssef Hospital Center	Clinical isolates (72)	CRPa	66 (91.7)	58.3	91.7	0	34.8	65.3	-	[59]
2018- 2019	Retrospective	Saydet Zgharta University Medical Center	LTRI (12) Urine (3) Wound (5)	CRPa	20 (100)	100	100	0	0	85	*bla*_VIM-2_ (16) Mutation in the *oprD* gene (18)	[60]
2013	Cross-sectional	Different farms	Stool of livestock animals (4)	CRPa	4 (100)	100	100	0	100	75	*bla*_VIM-2_ (4)	[38]
**South**												
2006–2012	Prospective	South (urban and rural hospitals)	Wounds of injured victims from sub-munition explosions (18)	RI	18 (0)	6	0	-	6	11	-	[61]

CRPa, Carbapenem Resistant *Pseudomonas aeruginosa*; RI, Random *Pseudomonas aeruginosa* isolates, LRT, Lower respiratory tract infections; ENT, Ear, nose, and throat; UTIs, Urinary tract infections*;* ICU, Intensive care unit; CAZ, Ceftazidime; IMP, Imipenem; CST, Colistin; AMK, Amikacin; LVX, Levofloxacin; -, Not determined. ^!^ Meropenem was reported if imipenem was not available. ^§^ Ciprofloxacin was reported if levofloxacin was not available.

## Data Availability

Not applicable.

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
