# Peer review of "The Emergence and Dissemination of Multidrug Resistant *Pseudomonas aeruginosa* in Lebanon: Current Status and Challenges during the Economic Crisis"

_antibiotics, 2022, doi:10.3390/antibiotics11050687_

Round 1

Reviewer 1 Report

This review is focused on the literature on multidrug-resistant (MDR) P. aeruginosa isolated from humans (nosocomial and community), animals, and the environment in  Lebanon. The review has included the published literature and compiled them nicely in the table. However, I have a few suggestions which should be considered for improvement.

  1. The manuscript needs thorough proofreading for grammar, sentence formation, and use of repetitive words. I had a very hard time reading the manuscript.
  2. In my opinion, the author should include a figure for section 2. “Mechanisms of antimicrobial resistance in Pseudomonas aeruginosa” based on general literature.

Author Response

Reviewer 1

Comment 1. The manuscript needs thorough proofreading for grammar, sentence formation, and use of repetitive words. I had a very hard time reading the manuscript.

- The paper has been proofread by at least two native English speakers prior to submission. Also, Dr. Kassem has native English language proficiency. The minor errors that were pointed to by the reviewer were largely due to formatting issues (transfer to MDPI template and accepting track changes). We are grateful to the reviewer for pointing these out. To comply with the reviewer, we have fixed these minor errors. Furthermore, we carefully check every sentence and simplified those that we thought were a bit unclear. Please check the edits highlighted ion the manuscript. Also, please note that the other two reviewers deemed the language to be suitable.

Comment 2. In my opinion, the author should include a figure for section 2. “Mechanisms of antimicrobial resistance in Pseudomonas aeruginosa” based on general literature.

- We thank the reviewer for this comment. We added a figure that shows resistance in P. aeruginosa as requested by the reviewer. Please check Figure 1.

Reviewer 2 Report

I have very few comments, overall the manuscript is well written and data from developing countries are important

Abstract: provide details:

 Our analysis showed that the bacterium was predominant in lesions of patients on mechanical ventilation (n= xx) and in burn patients and those with diabetic foot infections (n= xx) and hematological malignancies (n= xx). We also found that carbapenem-resistance in P. aeruginosa isolates in Lebanon in-26 volved both enzymatic and non-enzymatic mechanisms

Line 226: P. aeruginosa with a different font?

Lines 228-231: need a reference

Author Response

Reviewer 2

Comment 1. Abstract: provide details:

Our analysis showed that the bacterium was predominant in lesions of patients on mechanical ventilation (n= xx) and in burn patients and those with diabetic foot infections (n= xx) and hematological malignancies (n= xx). We also found that carbapenem-resistance in P. aeruginosa isolates in Lebanon in-26 volved both enzymatic and non-enzymatic mechanisms.

- We thank the reviewer for this comment. We have chosen to list these numbers in the table (Table 1), because they are generated by multiple centers over different or overlapping years. Therefore, combining the numbers might be misleading. Since all the numbers are detailed in the table, it would also be repetitive to include them in the abstract.  However, to comply with the reviewer, we modified the abstract by adding % of VIM-production (which was possible without being misleading). “… enzymatic and non-enzymatic mechanisms but depended predominantly on VIM-2 production (40.7%).”

Comment 2. Line 226: P. aeruginosa with a different font?

-  This is corrected in the revised manuscript.

Comment 3. Lines 228-231: need a reference.

- A refence was added.

van Duin, D.; Bonomo, R.A. Ceftazidime/avibactam and ceftolozane/tazobactam: Second-generation b-lactam/b-lactamase inhibitor combinations. Clin. infect. dis. 2016, 63, 234-41.

Reviewer 3 Report

The manuscript of Dabboussi et al. reviews the current emergency and dissemination of MDR P. aeruginosa isolates in Lebanon. This study is important for reinforcement of AMR surveillance programs and national action plans to combat antimicrobial resistance in Lebanon.

Major comments:

  • I suggest a graphical representation of the antimicrobials consumption over the time considerer for this study. For example, it is mentioned that Colistin consumption increased 5x between 2010 and 2017 (page 3, line 125-127). This information for other important antimicrobials related to P. aeruginosa infections management will be interesting to present. If possible, analysis of the emerging resistance to the same antimicrobials in the same time frame will be interesting.

Minor comments:

  • Reference 10, 11: Please review if these citations are complete. If they are webpages, please add the link.
  • Page 3, section 2, line 112: Please remove dot after “bacterium”.
  • Page 6, section 4, line 255: It is missing the end of the phrase.

Author Response

Reviewer 3

Comment 1. I suggest a graphical representation of the antimicrobials consumption over the time considerer for this study. For example, it is mentioned that Colistin consumption increased 5x between 2010 and 2017 (page 3, line 125-127). This information for other important antimicrobials related to P. aeruginosa infections management will be interesting to present. If possible, analysis of the emerging resistance to the same antimicrobials in the same time frame will be interesting.

- This is a valuable comment. However, unfortunately, this information is unavailable for other antibiotics. We conducted an extensive on-year research survey to identify colistin consumption (published and reference included). The consumption data for other antibiotics will require a similar approach and a lot of time and funding. We do not have currently the ability to do this research. However, the data on antimicrobial consumption is not critical fore this review, because the data on the emergence of resistant P. aeruginosa were thoroughly analyzed and represented. 

Comment 2. Reference 10, 11: Please review if these citations are complete. If they are webpages, please add the link.

- This was done as recommended by the reviewer.

Comment 3. Page 3, section 2, line 112: Please remove dot after “bacterium”.

- This was done as recommended.

Comment 4. Page 6, section 4, line 255: It is missing the end of the phrase.

- Apparently, the word “countries” was cut during formatting the MDPI template.  This issue was resolved in the revised manuscript.

Round 2

Reviewer 1 Report

The manuscript have improved a lot I have no further comments.